# Tree seedling functional traits mediate plant-soil feedback survival responses across a gradient of light availability

Katherine E. A. Wood[ID]1,2*, Richard K. Kobe1,2, Inés Ibáñez3, Sarah McCarthy-Neumann1,4

1 Department of Forestry, Michigan State University, East Lansing, Michigan, United States of America, 2 Program in Ecology, Evolution and Behavior, Michigan State University, East Lansing, Michigan, United States of America, 3 School for Environment and Sustainability, University of Michigan, Ann Arbor, Michigan, United States of America, 4 Department of Agricultural and Environmental Sciences, Tennessee State University, Nashville, Tennessee, United States of America

* woodkat7@msu.edu

## Abstract

**1.** Though not often examined together, both plant-soil feedbacks (PSFs) and functional traits have important influences on plant community dynamics and could interact. For example, seedling functional traits could impact seedling survivorship responses to soils cultured by conspecific versus heterospecific adults. Furthermore, levels of functional traits could vary with soil culturing source. In addition, these relationships might shift with light availability, which can affect trait values, microbe abundance, and whether mycorrhizal colonization is mutualistic or parasitic to seedlings. **2.** To determine the extent to which functional traits mediate PSFs via seedling survival, we conducted a field experiment. We planted seedlings of four temperate tree species across a gradient of light availability and into soil cores collected beneath conspecific (sterilized and live) and heterospecific adults. We monitored seedling survival twice per week over one growing season, and we randomly selected subsets of seedlings to measure mycorrhizal colonization and phenolics, lignin, and NSC levels at three weeks. **3.** Though evidence for PSFs was limited, *Acer saccharum* seedlings exhibited positive PSFs (i.e., higher survival in conspecific than heterospecific soils). In addition, soil microbes had a negative effect on *A. saccharum* and *Prunus serotina* seedling survival, with reduced survival in live versus sterilized conspecific soil. In general, we found higher trait values (measured amounts of a given trait) in conspecific than heterospecific soils and higher light availability. Additionally, *A. saccharum* survival increased with higher levels of phenolics, which were higher in conspecific soils and high light. *Quercus alba* survival decreased with higher AMF colonization. **4.** We demonstrate that functional trait values in seedlings as young as three weeks vary in response to soil source and light availability. Moreover, seedling survivorship was associated with trait values for two species, despite both drought and heavy rainfall during the growing season that may have obscured survivorship-trait relationships. These results suggest that seedling traits could have an important role in mediating the effects of local soil source and light levels on seedling survivorship and thus plant traits could have an important role in PSFs.

**Data Availability Statement:** All data and analysis files relevant to this study are available at Dryad: https://doi.org/10.5061/dryad.xd2547dpw.

**Funding:** SMN, RKK, II. National Science Foundation (NSF DEB 145732), Michigan State University, Alma College. The funders had no role in study design, data collection and analysis, decision to publish, or preparation of the manuscript.

**Competing interests:** The authors have declared that no competing interests exist.

## Introduction

Though often examined separately, both plant-soil feedbacks (PSFs) and functional traits are important in plant community dynamics [1–5]. PSFs are a continuous feedback loop whereby plants modify properties of the soil they are growing in and influence the performance of future plants growing in that soil [6]. These feedbacks subsequently affect community composition, which in turn influences soil properties, and so on. The net effect of interactions results in positive (better performance in conspecific soils), negative (better performance in heterospecific soils), or neutral PSFs.

The putative agents of PSFs are soil-borne microbes, like mycorrhizae and pathogens [7, 8]. Arbuscular mycorrhizal fungi (AMF) are often mutualistic, exchanging water and nutrients for photosynthates [9]. Soil-borne pathogens, including fungi, oomycetes, and bacteria, can cause the death of entire seedling cohorts [10, 11], and pathogens with higher effective specialization are more abundant in conspecific soils [12, 13]. Mycorrhizal colonization is frequently higher in conspecific soils and in soils cultured by adult trees of the same mycorrhizal type [14–16]. These soils contain mycorrhizal genotypes that are well-suited to colonizing the adult trees growing in them [5, 17].

Functional traits are measurable morphological or physiological attributes affecting plant performance [18] that can translate into impacts on community dynamics. Despite the important role of plant survival in PSF [19, 20], traits promoting faster growth (e.g., specific leaf area, specific root length, height) have been the focus of most PSF studies [21–23]. Frequently, defensive traits are accounted for by assuming that species with fast growth rates have low investment in defense, and vice-versa [22, 23]. However, tree seedling survivorship is likely to have greater effects on future community dynamics and composition than growth [24]. Thus, while little studied, functional traits that influence tree seedling survivorship in response to PSFs could be a crucial mechanism governing seedling and forest community dynamics.

Plant functional traits could influence PSFs and vice-versa [23, 25, 26]. Functional traits that influence plant defense against and recovery from attack by soil-borne microbes include phenolics, lignin, and nonstructural carbohydrates (NSC). Phenolics and lignin can serve as chemical [27] and physical [28] defenses against soil-borne microbes. NSC can be mobilized to repair damaged tissues [29]. Additionally, percent root colonization by mycorrhizae can be treated as a trait [30] conferring defense against pathogens [14]. AMF can provide indirect defense against pathogens by competing for space on plant roots [31], and EMF can provide direct defense by forming a protective physical sheath on young roots [32]. Also, both AMF and EMF can increase their host plant's resource acquisition, which can be allocated to defensive and recovery traits [33, 34].

Phenolics, lignin, and NSC are likely affected by soil source (e.g., conspecific versus heterospecific soil). Phenolics production can be induced by mycorrhizal colonization [35] and potentially by fungal pathogens [36]. We expect that phenolics production should subsequently be higher in conspecific soils, where there should be higher colonization by mycorrhizal fungi and infection by effectively-specialized soil-borne pathogens [12, 13]. It is unclear whether lignin production is influenced by conspecific soils. However, it could be driven by soil nutrient availability, which can be impacted by microbes [37, 38]. NSC should be lower in conspecific soils, due to greater resource allocation to symbionts [39] and recovery against pathogen infection [40, 41].

Both PSFs and functional traits can shift across environmental gradients like light availability [42, 43]. However, most studies have not integrated abiotic factors when evaluating both traits and PSFs [22]. Shifts in light can change microbial composition and abundance [44, 45], which may alter seedlings' ability to defend against or recover from disease. AMF are more

abundant in higher light [44, 46, 47]; however, in low light they can act parasitically and thereby decrease seedling survival [48]. Higher mortality from pathogens typically occurs in low light [20, 42], where wetter and cooler conditions enhance microbe reproduction and dispersal [13, 45, 49]. Light availability can also modify functional trait level, including reduced production of phenolics [27] and lignin [50, 51]. Additionally, carbon limitation in shade and lower stored nonstructural carbohydrates (NSC) may constrain recovery from disease [52, 53].

Our overall conceptual framework (Fig 1) is that soil source and light availability influence trait levels, which in turn influence tree seedling survival. Thus, plant traits have an important role in mediating PSFs. We hypothesized that:

1. Negative PSFs are widespread across tree species and are more prevalent under low than high light. Furthermore, these differences are only present with soil-borne microbes are present. This result would indicate that soil-borne microbes drive negative PSFs in low light availability directly through increased pathogen abundance and/or a shift from positive to

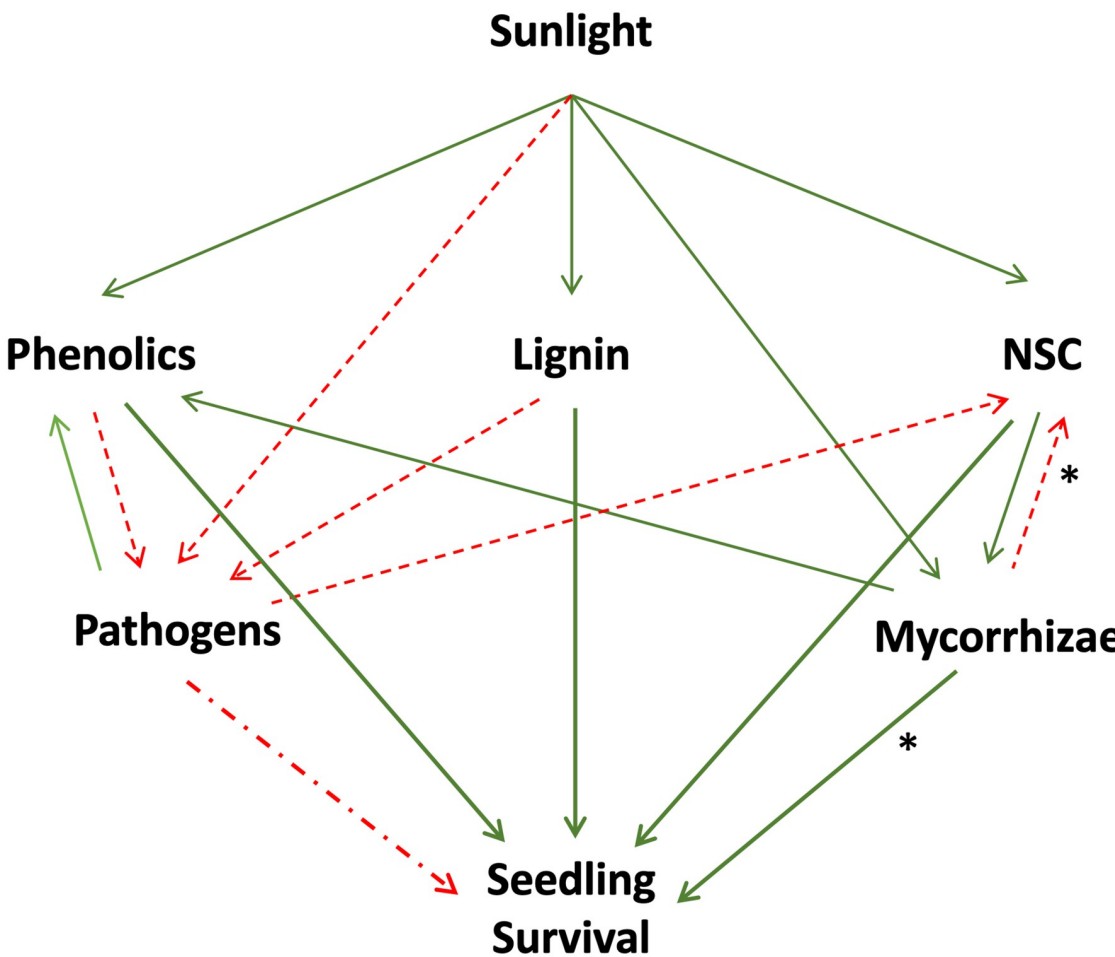

**Fig 1. Conceptual diagram demonstrating the relationships between light availability, functional traits (phenolics, lignin, and nonstructural carbohydrates [NSC]), colonization by mycorrhizal fungi, and tree seedling survival.** Green, solid lines indicate a positive relationship. Red, dashed lines indicate a negative relationship. Lines that directly influence tree seedling survival are thicker. Stars (*) next to the lines linking 'Mycorrhizae' with 'NSC' and 'Seedling Survival' indicate that this relationship is usually positive but can shift to neutral or negative.

negative in the plant-mycorrhizal fungi relationship, and/or indirectly through decreased levels of defensive traits.

2. Mycorrhizal colonization is greater in conspecific soils and in higher light. This result would indicate that mycorrhizal colonization is promoted by effectively-specialized microbes in conspecific soils and greater resource availability (e.g., NSC) in high light.

3. The defensive functional traits phenolics and lignin are induced to higher levels in soils cultured by conspecific adults and in high light availability. This result would indicate that defensive functional trait production is driven by the presence of effectively-specialized parasitic microbes expected in conspecific soils and by greater carbon income expected in higher light.

4. The recovery trait NSC is lower in soils cultured by conspecific adults and in low light availability. This result would indicate that NSC is drawn down in the presence of effectively-specialized parasitic microbes expected in conspecific soils, in addition to lower carbon income relative to use expected in lower light.

5. Finally, we hypothesized that seedling survival increases as mycorrhizal colonization, phenolics, lignin, and NSC also increase. This result would indicate that PSFs can, in part, be mediated by the degree of mycorrhizal colonization and changes in functional trait values responding to variation among soil types and in light availability.

## Methods

We conducted a factorial blocked design field experiment, consisting of four tree species, seven soil sources (sterilized conspecific, live conspecific, and five heterospecific), and a gradient of forest understory light levels (low, medium, and high), for a total of 3,024 seedlings. We monitored seedling survival twice per week over one growing season, and we randomly selected subsets of seedlings to measure mycorrhizal colonization and phenolics, lignin, and NSC measurements at three weeks. We used Cox proportional hazards survival models to evaluate survival and linear mixed effects models to test how light availability and soil source influence traits.

### Study location

The research site is a 100 ha mixed hardwood forest stand in mid-Michigan, at Alma College's Ecological Field Station (43˚23'32.0"N, 84˚53'41.5"W). Alma College granted permission to undertake this research and collect plant and soil materials; a formal field permit was not required. This forest has not been logged since 1897 and lies in an ecological tension zone between northern coniferous and southern deciduous forests. The dominant species in this forest is sugar maple (*Acer saccharum*), a shade-tolerant canopy tree species. Other common trees in the forest include red maple (*A. rubrum*) and big-toothed aspen (*Populus grandidentata*).

**Species selection.** We identified adult trees for soil collection and established field plots in a 3-ha mapped section of the forest. We initially chose six tree species native to the research site: red maple (*A. rubrum*), sugar maple (*A. saccharum*), big-toothed aspen (*P. grandidentata*), black cherry (*Prunus serotina*), white oak (*Quercus alba*), and northern red oak (*Q. rubra*). *A. rubrum* and *P. grandidentata* seedlings experienced high (> 80%) mortality within two weeks of planting, suggesting poor seed source or propagation methods. Thus, while still included as soil sources, they were not included in analyses of seedling survival, mycorrhizal colonization, or functional traits.

## Soil sources and planting

We collected intact soil cores from May to July 2016 and April to May 2017 (S1 Table). To minimize potential for multispecies culturing of soil, we took soil cores under trees that were at least two crown diameters away from adults of other species. Using a custom-made mechanized soil core sampler (Giddings Machine Co; Windsor, CO, USA), we removed intact soil cores (9 cm diameter and 30 cm deep for planted *A. saccharum* and *P. serotina* seedlings, or 46 cm long for *Q. alba* and *Q. rubra* seedlings) within 1 m from the bole of six mature randomly selected adults for each of the six study species (36 trees total). We maintained soil cores from each adult as separate replicates [54, 55].

Intact soil cores with plastic liners were converted into pots by drilling two 7.5 cm diameter holes into the sides and adhering a 0.5 μm nylon mesh covering over side holes and the bottom opening. Such pots are an established method for studying common mycorrhizal networks in forests [58–60]. The mesh prevents roots, fungal hyphae, oomycetes and pathogenic fungi from passing in or out, with minimal effect on water and nutrient flows [61]. We did not use multi-stage greenhouse culturing [7], because in-situ natural culturing already had occurred for these long-lived trees and should more closely characterize PSFs occurring in the field.

Soil samples for the sterilized conspecific soil treatment were exposed to gamma irradiation (30–70 kGy; Sterigenics International, Schaumburg, IL, USA) in July 2017 and allowed to rest for at least one month to minimize post-sterilization nutrient spikes. Gamma irradiation is highly effective at killing soil microorganisms and typically has minimal effects on soil chemical and physical properties [62]. Nevertheless, we tested the sterilized versus live soils using plant root simulator (PRS™) probes (Western Ag Innovations Inc., Saskatchewan, Canada) and found no effect of sterilization on soil nutrient availability (S3 and S4 Tables; S3 Fig).

After resting, pots were transplanted into eighteen 8.4 m x 6.6 m common-garden field plots that fell within three general light groupings (low, medium, and high). Existing vegetation and leaves in each plot were removed to reduce light interception. We then took precise measurements of light availability by analyzing canopy photos with HemiView software (Delta-T Devices, Ltd., Burwell, England; S5 Fig).

We planted 108 seedling pots per species × soil source, evenly distributed across the 18 field plots. A single surface-sterilized seed with a newly-emerged radicle was planted into each pot. Seeds for *Q. alba* were purchased from Sheffields Seed Co (Locke, NY, USA) and seeds for all other species were collected from mid-Michigan forests. Variation among seed source populations in survivorship, mycorrhizal colonization, and functional traits was likely minimal [63]. In June 2018, one week prior to planting, we added 1 cm of a 1:1 mixture of peat moss and fresh or sterilized soil to increase transplant success and provide fresh inoculum. In a previous trial run, we found that seedlings planted with peat moss and fresh soil had reduced transplant shock (authors et al. personal observation).

To minimize disease from non-experimental soil sources, seeds were surface sterilized with 0.6% NaOCl solution prior to stratification and prior to germination. To avoid cross-contamination, all tools and surfaces that were exposed to soil were soaked in 10% bleach or surface sprayed with 70% EtOH and then rinsed with deionized water. To minimize browsing and digging-up of seedlings by vertebrates, we erected galvanized hardware cloth (6 x 6 cm openings) to 1.8 m height around each plot. We also glued hardware cloth with 0.25 cm x 0.25 cm openings to the top of each pot. Seedlings likely did not experience significant shading due to the addition of the hardware cloth and often grew above the cloth within 2 weeks of planting.

## Survival and functional traits

We censused seedling survival twice per week for 16 weeks. Mortality at the first two censuses after planting were attributed to transplant shock or poor seed source; these seedlings were not used in subsequent analyses and pots were re-planted with the same seedling species.

Three weeks after planting, we harvested six seedlings per treatment combination to measure mycorrhizal colonization, phenolics, lignin, and NSC. We chose this harvest date since, in a previous greenhouse experiment, mortality curves for tree seedlings subjected to soil-borne pathogens often increased at week three and peaked between four to six weeks after germination [63]. For measurements, we used established protocols: AMF and EMF colonization [64, 65], phenolics [66, 67], lignin (ANKOM Technologies, Macedon, NY, USA), and NSC [68]. Due to the small size of three-week-old seedlings and the destructive nature of each measurement, half of the harvested seedlings were allocated to measurement of NSC (stem and root), and half of the seedlings were allocated to measurement of phenolics (hypocotyl), lignin (stem), and mycorrhizal colonization (roots).

## Statistical analysis

To evaluate hypotheses 1, we analyzed seedling survival over 16 weeks with Cox proportional hazards regression [69]. We ran species-specific models, using soil source and light availability as fixed effects, and plot and adult tree as random effects. The best fitting models for seedling survival did not include any interactions. We compared survival in live conspecific versus heterospecific soils [23, 70]. Greater survival in live conspecific than heterospecific soils indicated positive PSFs. We also compared survival in sterilized versus live conspecific soils. Higher survival in sterilized than live conspecific soils indicated that microbes influenced PSFs.

To evaluate hypotheses 2–4, we analyzed measured amounts of mycorrhizal colonization, phenolics, lignin, and NSC with linear mixed effects models. We ran species-specific models for each trait, using soil source and light availability as fixed effects, and plot and adult tree as random effects. We used a priori contrasts to compare levels of measured traits in live conspecific versus heterospecific soils and to compare levels of measured traits in sterilized versus live conspecific soils.

To evaluate hypothesis 5, we analyzed seedling survival over 16 weeks with Cox proportional hazards regression [69]. We ran species-specific models, with mycorrhizal colonization, phenolics, lignin, and NSC as fixed effects. We imputed colonization and trait data from seedlings harvested at three weeks, for each combination of seedling species, plot, soil source, and light level. We accounted for possible collinearity between traits by calculating variance inflation factors (VIF) for each model and removing variables with VIF > 5. For *A. saccharum*, *A. rubrum*, and *Q. rubra*, NSC was removed from the final models, and for *Q. alba*, lignin was removed from the final models. NSC was highly correlated with lignin for all study species and with phenolics for *Q. rubra* (S8 Fig).

For all models, light availability was first evaluated as a continuous variable, using Indirect Site Factor (ISF) quantified with canopy photos. ISF represents the proportion of diffuse (indirect) solar radiation reaching a given location, relative to an open site and was calculated using HemiView software (Delta-T Devices, Ltd., Burwell, England). For post-hoc comparisons and figures, we divided seedlings according to light group, splitting the range of light availability into three bins (low = 0.032–0.075 ISF, medium = 0.075–0.118 ISF, and high = 0.118–0.161 ISF). These light thresholds were determined by dividing the range of light availability across the field plots into three bins. Heterospecific soils were modeled as both pooled and unpooled/specific soils; when evaluating post-hoc comparisons, we used pooled heterospecific soils.

All analyses were performed with R version 3.5.1 [71]. We used the lme4 package [72] to evaluate linear models. We used the "coxph" function in the survival package [73] to fit Cox proportional hazards regression models. We tested the significance of main effects using a likelihood ratio test with the "Anova" function. We tested for multicollinearity variance inflation factors using the "vif" function in the car package [74]. Post-hoc Tukey pairwise comparisons of significant main effects and Bonferonni corrections for multiple comparisons were made using the "emmeans" function in the multcomp package [75, 76]. We used the missForest package [77] to impute trait data for seedlings monitored for survival.

## Results

### Negative PSFs were not widespread among tree species, nor were they more prevalent in low light availability

No species experienced negative PSFs (defined as lower survival in conspecific versus heterospecific soils). However, *A. saccharum* experienced positive PSFs with higher survival in conspecific than pooled heterospecific soil (LR$\chi^2$ = 8.60, *p* < 0.01; Fig 2A, Table 1A). Seedling survival was lower in live than sterilized conspecific soil for both *A. saccharum* (LR$\chi^2$ = 61.78, *p* < 0.01) and *P. serotina* (LR$\chi^2$ = 1.52, *p* < 0.01), suggesting an effect of soil-borne microbes. Although there was a positive effect of light on survival for *P. serotina* (LR$\chi^2$ = 4.09, *p* = 0.04) and *Q. rubra* (LR$\chi^2$ = 9.02, *p* < 0.01; Fig 2B, Table 1B) there was no significant interaction between soil source and light availability for any of the models with pooled heterospecific soils. When heterospecific soils were not pooled, there was a significant interaction between light and soil source, but only for *P. serotina* (LR$\chi^2$ = 6.860, *p* = 0.009). Thus, our expectation that negative PSFs are widespread among species and are more prevalent in low light availability was not supported.

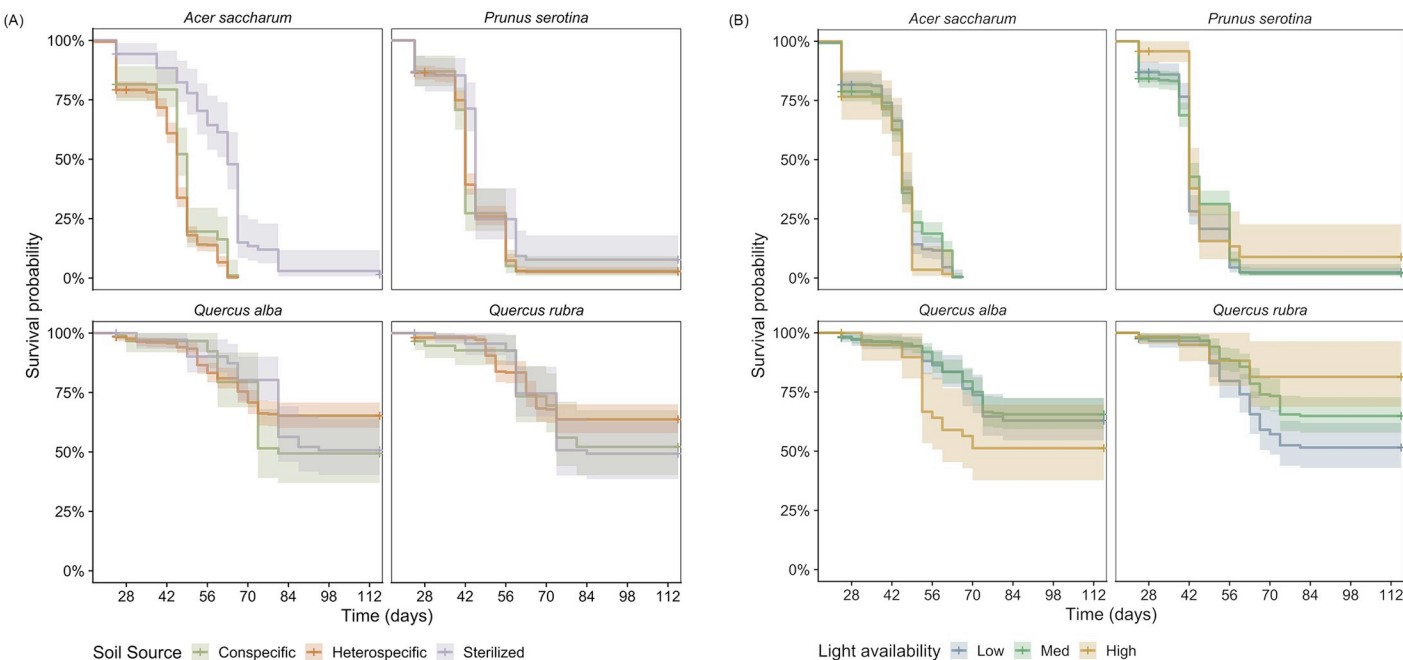

**Fig 2. Kaplan-Meier survival plots.** Effects of A) soil source (conspecific, pooled heterospecific, and sterilized conspecific) and B) light availability on seedling survival. For visualization, light availability was binned into 3 levels: Low = 0.032–0.075 ISF, Med = 0.075–0.118 ISF, and High = 0.118–0.161 ISF. Shaded regions indicate 95% confidence intervals about the mean.

**Table 1. Number of surviving seedlings at the end of the growing season.** Data is presented for each A) species × soil source and B) species × light level as both an absolute number and percentage. Because there was not a significant interaction between soil source and light availability on seedling survival, they are presented separately, corresponding with Fig 2A and 2B. Soil sources include sterilized conspecific, live conspecific, and pooled heterospecific soils. Light availability was binned into 3 levels: Low = 0.032–0.075 ISF, Med = 0.075–0.118 ISF, and High = 0.118–0.161 ISF.

| Species | A) Soil source | | | B) Light availability | | |
|---|---|---|---|---|---|---|
| | Sterilized conspecific | Live conspecific | Hetero-specific | Low | Med | High |
| *Acer saccharum* | 1 (1%) | 0 (0%) | 0 (0%) | 0 (0%) | 1 (0.2%) | 0 (0%) |
| *Prunus serotina* | 5 (8.9%) | 3 (3.3%) | 12 (2.8%) | 8 (3.9%) | 7 (2.3%) | 5 (7.7%) |
| *Quercus alba* | 36 (50.7%) | 23 (50%) | 199 (66.3%) | 83 (60.1%) | 150 (64.7%) | 25 (51.1%) |
| *Quercus rubra* | 33 (49.3%) | 27 (54%) | 152 (65%) | 66 (52%) | 118 (62.8%) | 28 (77.8%) |

## Mycorrhizal colonization and seedling functional traits varied across both soil source and light availability

AMF colonization was 11% higher in conspecific than pooled heterospecific soil only for *A. saccharum* ($t_{2344}$ = 1.84, marginally significant at $p$ = 0.07; Fig 3A). For the other study species, AMF colonization was higher in pooled heterospecific than conspecific soils: 12% for *P. serotina* ($t_{2344}$ = 3.88, $p$ < 0.01), 16% for *Q. alba* ($t_{2344}$ = 2.38, $p$ = 0.02), and 12% for *Q. rubra* ($t_{2344}$ = 2.05, $p$ = 0.04). As predicted, AMF colonization increased with light for *P. serotina* (slope = 108% / ISF, $F_{1,2344}$ = 35, $p$ < 0.01) and *Q. rubra* (slope = 53% / ISF, $F_{1,2344}$ = 6.42, $p$ < 0.01), but not for *A. saccharum*. Contrary to our expectations, AMF colonization decreased with light for *Q. alba*, which is primarily associated with EMF (slope = -49% / ISF, $F_{1,2344}$ = 6.08, $p$ = 0.01).

EMF colonization was higher in conspecific than pooled heterospecific soil by 16% for *Q. alba* across all light levels ($t_{1063}$ = 2.72, $p$ = 0.01; Fig 3B) and by 22% for *Q. rubra* in high, but not low light ($t_{1063}$ = 4.74, $p$ < 0.01). EMF colonization increased with light availability for *Q. rubra* (slope = 264.2% / ISF, $F_{1,1063}$ = 63.02, $p$ < 0.01), especially in conspecific soil (slope = 413% / ISF).

Phenolics (nmol Gallic acid equivalents per mg dry extract) were higher in live than sterilized conspecific soils for *A. saccharum* (227%, $t_{793}$ = 7.85, $p$ < 0.001), *P. serotina* (173%, $t_{793}$ = 6.77, $p$ < 0.001), and *Q. alba* (51.7%, $t_{793}$ = 43.73, $p$ < 0.001). As expected, phenolics were higher in conspecific than pooled heterospecific soil for *A. saccharum* (23%, $t_{2344}$ = 10.56, $p$ < 0.01; Fig 4A) and *Q. alba* (4%, $t_{2344}$ = 4.44, $p$ < 0.01). Conversely, phenolics were 69% higher in pooled heterospecific soil for *P. serotina* ($t_{2344}$ = 6.96, $p$ < 0.01). For *Q. rubra*, phenolics were 18% higher in conspecific soil at high light ($t_{2344}$ = 5.89, $p$ < 0.01) and 29% higher in pooled heterospecific soil at low light ($t_{2344}$ = 12.77, $p$ < 0.01). Phenolics increased with light availability for all four study species (*A. saccharum*: slope = 5.70 nmol / ISF, $F_{1,2344}$ = 64.7, $p$ < 0.01; *P. serotina*: slope = 4.15 nmol / ISF, $F_{1,2344}$ = 34.48, $p$ < 0.001; *Q. alba*: slope = 10.51 nmol / ISF, $F_{1,2344}$ = 187.07, $p$ < 0.01; *Q. rubra*: slope = 12.73 nmol / ISF, $F_{1,2344}$ = 249.42, $p$ < 0.01). For *Q. rubra*, this trend appeared to be driven by conspecific soil (slope = 23.26 nmol / ISF).

Percent dry mass lignin was higher in conspecific than pooled heterospecific soil by 11% for *Q. alba* ($t_{2344}$ = 8.60, $p$ < 0.01) and 5.8% for *Q. rubra* ($t_{2344}$ = 5.61, $p$ < 0.01), across all light levels (Fig 4B). For both *A. saccharum* and *P. serotina*, lignin did not vary between conspecific and pooled heterospecific soil. Lignin increased with light availability for *A. saccharum* (slope = 38% / ISF, $F_{1,2344}$ = 82.34, $p$ < 0.01) and *P. serotina* (slope = 57% / ISF, $F_{1,2344}$ = 184.51, $p$ < 0.01). There was no effect of light on lignin for *Q. alba*. Contrary to our predictions, for *Q. rubra*, lignin decreased with light availability (slope = -22% / ISF, $F_{1,2344}$ = 21.42,

## A) Percent colonization by AMF

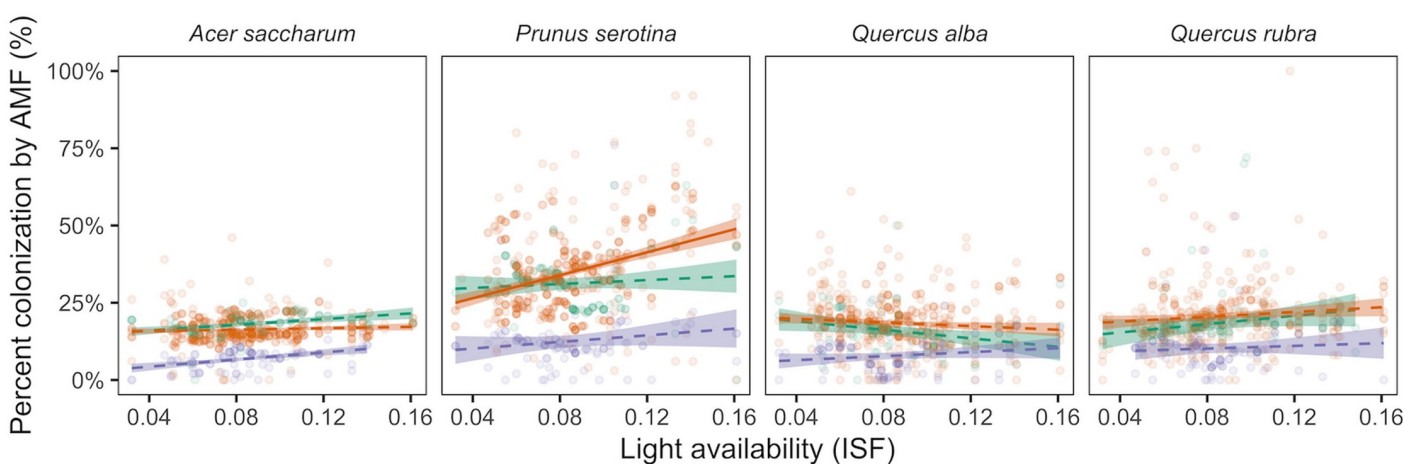

## B) Percent colonization by EMF

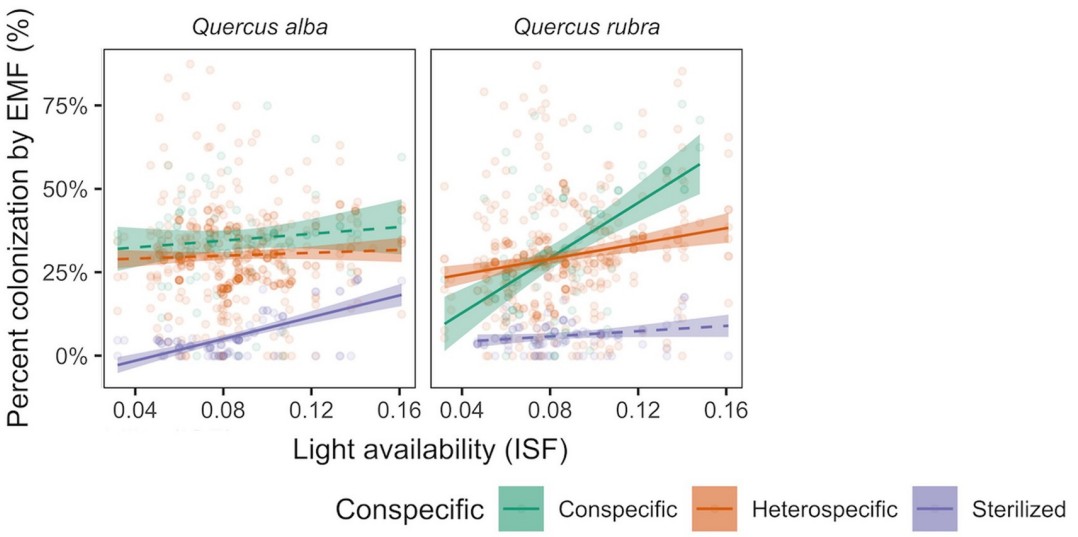

**Fig 3. Effect of soil source and light availability on percent mycorrhizal colonization.** By A) AMF and B) EMF (only *Q. alba* and *Q. rubra* are colonized by EMF). Shaded regions indicate 95% confidence intervals about the mean. Solid lines have a slope significantly different from zero ($p < 0.05$).

$p < 0.01$); this trend appeared to be driven by conspecific soil (slope = -40% / ISF). Indicating a potential effect of soil biota, lignin (percent dry mass) was higher in live than sterilized conspecific soils for all four study species: *A. saccharum* (12%, $t_{793} = 6.40$, $p < 0.001$), *P. serotina* (13%, $t_{793} = 4.18$, $p < 0.001$), *Q. alba* (12%, $t_{793} = 10.56$, $p < 0.001$), and *Q. rubra* (2.4%, $t_{793} = 2.46$, $p = 0.014$).

Percent dry mass NSC was higher in pooled heterospecific than conspecific soil across all light levels for *Q. alba* (15%, $t_{2344} = 9.96$, $p < 0.01$; Fig 4C). For *P. serotina* and *Q. rubra*, NSC was higher in conspecific soil at low light (*P. serotina*: 1.9%, $t_{2344} = 3.41$, $p < 0.01$; *Q. rubra*: 13%, $t_{2344} = 9.34$, $p < 0.01$), but did not vary with soil source at high light. For all four study species, NSC increased with light availability (*A. saccharum*: slope = 21% / ISF, $F_{1,2344} = 22.05$, $p < 0.01$; *P. serotina*: slope = 75% / ISF, $F_{1,2344} = 287.75$, $p < 0.01$; *Q. alba*: slope = 19% / ISF,

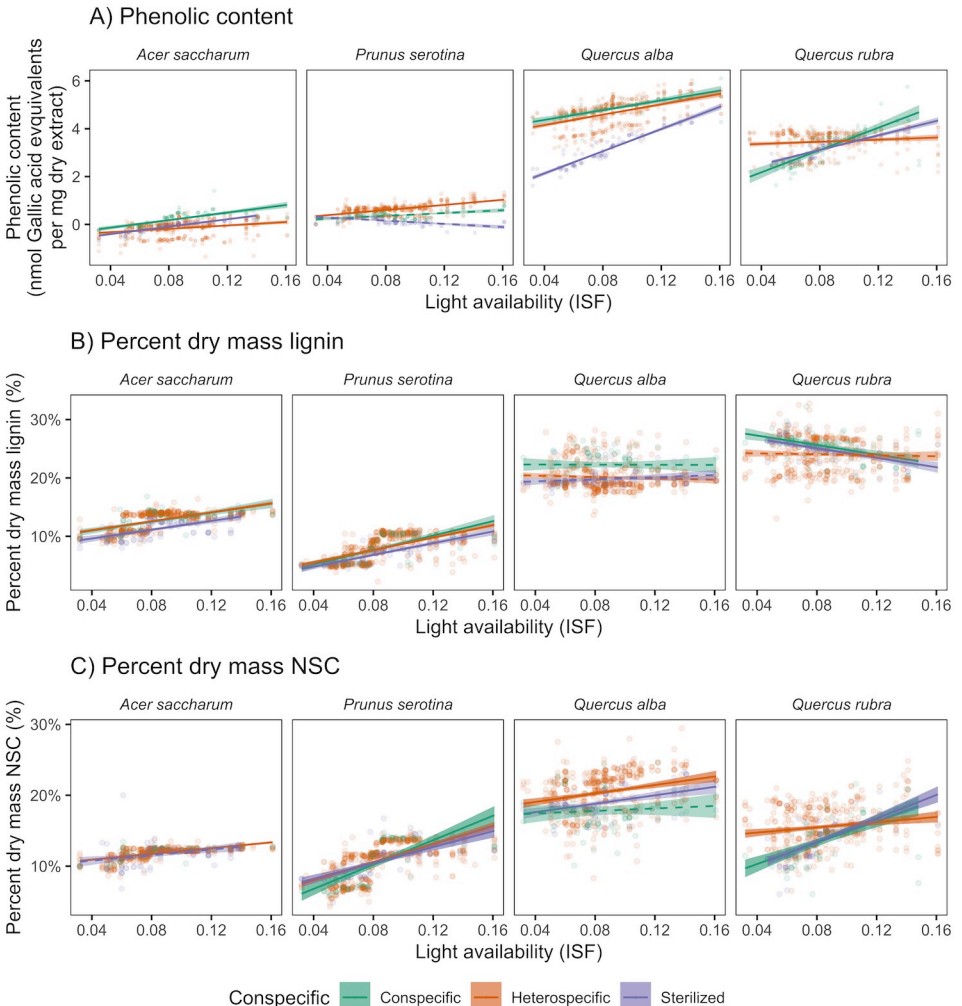

**Fig 4. Effect of soil source and light availability on functional traits.** Traits include: A) phenolics (nmol Gallic acid equivalents per mg dry mass), B) percent dry mass lignin, and C) percent dry mass NSC. Some lines are truncated, because not enough seedlings survived in the lowest or highest light levels. Shaded regions indicate 95% confidence intervals about the mean. Solid lines have a slope significantly different from zero ($p < 0.05$).

$F_{1,2344} = 16.39$, $p < 0.01$; *Q. rubra*: slope = 47% / ISF, $F_{1,2344} = 87.42$, $p < 0.01$). This trend appeared to be driven by pooled heterospecific soil for *Q. alba* (slope = 30), and conspecific soil for *Q. rubra* (slope = 76% / ISF). NSC was higher in sterilized than live conspecific soils for *Q. alba* (6.7%, $t_{793} = 4.17$, $p < 0.001$).

## Mycorrhizal colonization and functional traits had limited effects on seedling survival

From the Cox survival models, we interpreted hazard ratios (HR), an integration of the hazard experienced by seedlings across the study duration. HR < 1 indicates decreased hazard relative to the baseline (i.e., increased survival); HR > 1 indicates increased hazard (i.e., decreased survival). Traits predicted survival for two species: phenolics had a positive effect on survival for *A. saccharum* (HR = 0.73, LR$\chi^2$ = 4.20, $p = 0.04$; Fig 5) and AMF colonization had a negative effect for *Q. alba* (HR = 1.04, LR$\chi^2$ = 4.18, $p = 0.04$).

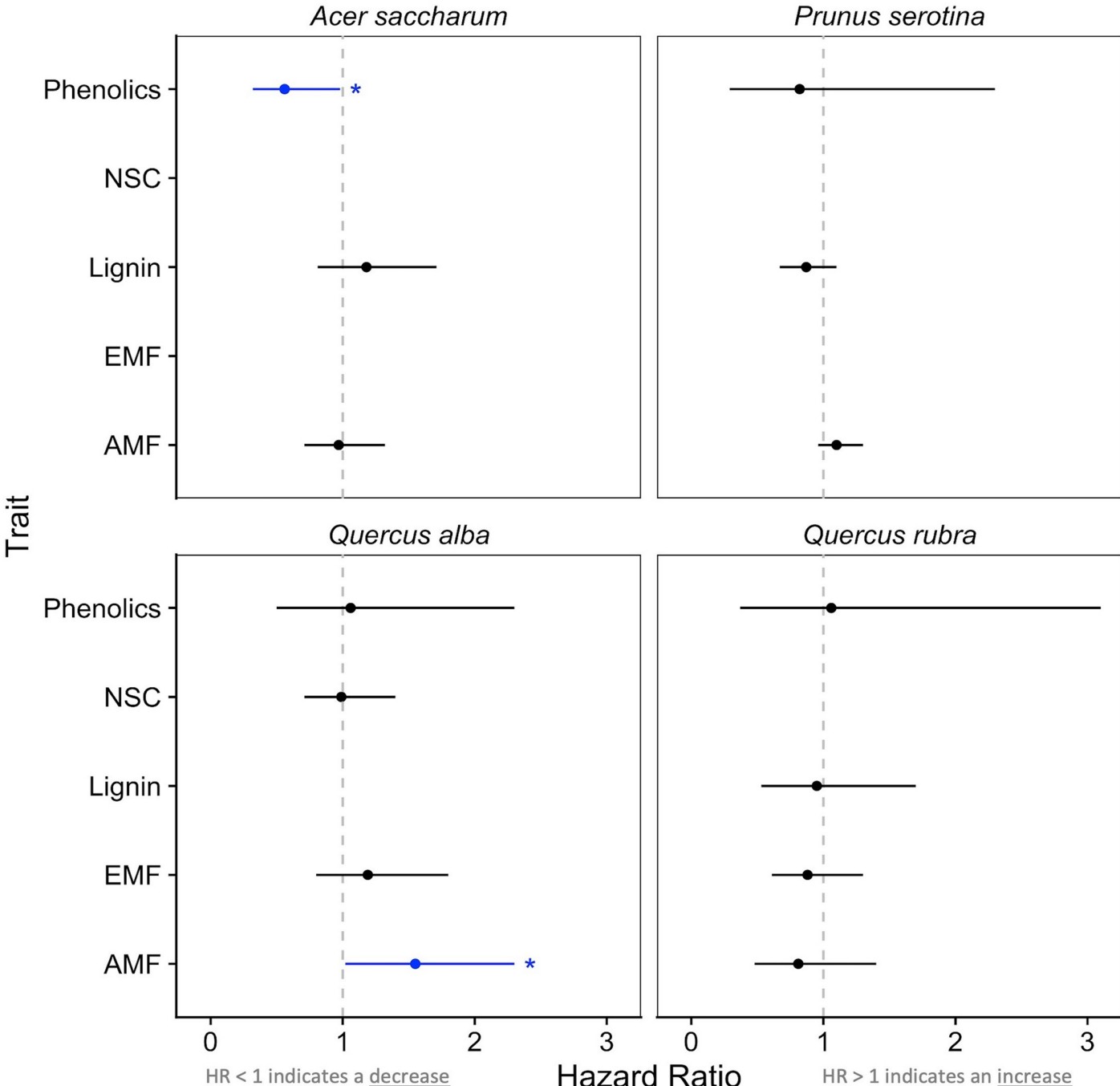

**Fig 5. Hazard ratios (HR) demonstrating the effect of mycorrhizal colonization and functional traits on seedling survival over the growing season.** HR > 1 indicates an increase in mortality and HR < 1 indicates a decrease in mortality as the trait increases. Statistically-significant effects ($p < 0.05$) are colored blue. Species × trait combinations that are blank were removed from the final models due to high collinearity (VIF > 5).

## Discussion

While we found no evidence of PSFs in this field study, functional traits varied in response to soil source and light availability in seedlings as young as three weeks old. We also found limited associations between functional traits and seedling survivorship. Our results support that

while seedling functional traits are very responsive to soil source and light availability, PSFs may be less prevalent in field conditions where multiple environmental factors influence seedling survivorship.

## PSFs were not widespread among species, nor were they more prevalent in low light availability

Only one of four study species experienced PSFs between conspecific and heterospecific soils, and there were very few interactions between light availability and soil source on seedling survival. We expected to find stronger negative PSFs in low light conditions [42], due to both greater limitation of light availability and higher prevalence of soil-borne pathogens.

Both *A. saccharum* and *P. serotina* experienced lower survival in live than sterilized conspecific soils, indicating that soil-borne microbes have negative effects on seedling survival. For *P. serotina*, although soil-borne microbes cultured in conspecific soils may have a negative effect on survival, the net effect of PSFs (assessed as survivorship in conspecific versus pooled heterospecific soils) appeared to be neutral [but see 78–80].

*A. saccharum* seedlings experienced net positive PSFs, having greater survival in conspecific than heterospecific soils. However, they had even greater survival in sterilized than live conspecific soils, consistent with McCarthy-Neumann and Ibáñez [42] and suggesting net negative effects of soil-borne microbes. There are at least three mutually compatible explanations for these results: 1) mutualistic microbes may provide greater benefit in conspecific than heterospecific soils; 2) there may be a greater negative effect of harmful microbes in heterospecific soils; or, 3) there may be unmeasured, more favorable abiotic effects in soils modified by *A. saccharum* adults in comparison to heterospecific soils [63].

Comparisons of PSFs in this study are presented as differences in rates (calculated as LR$\chi^2$), rather than differences in total number of surviving seedlings at the end of the growing season (Fig 2, Table 1). Both *A. saccharum* and *P. serotina* had zero (or near-zero) survival by the end of the growing season. However, investigating the environmental conditions that influence mortality rates for these seedlings is still meaningful for understanding forest communities. When adult *A. saccharum* and *P. serotina* trees produce thousands of seeds in a single growing season [57], small differences in survival rates can scale up to meaningful impacts on community composition over longer time periods.

## Mycorrhizal colonization and functional traits varied across both soil source and light availability

Our results demonstrate that tree seedlings, even as young as three weeks old, express intraspecific variation in mycorrhizal colonization and functional trait values, in response to conspecific versus heterospecific soil source and light availability. AMF colonization was higher in heterospecific than conspecific soil for most of the measured species, with the largest difference being for *P. serotina* at high light availability. This was in contrast to a study evaluating PSF in temperate tree species across North America, which found that AMF colonization was equal or greater in conspecific relative to heterospecific soils [14]. Our result may be because AMF are more generalized in host associations than EMF for our study species [81]. AMF colonization was highest in *P. grandidentata* soil for *P. serotina* and *Q. alba* seedlings, and in *A. saccharum* soil for *P. serotina* and *Q. rubra*, suggesting that AMF from *P. grandidentata* and *A. saccharum* soils can readily colonize multiple seedling species. Surprisingly, AMF colonization increased with light availability only for *P. serotina*. Across species, *P. serotina* also had the highest total AMF colonization. We speculate that, as a shade intolerant species, *P. serotina*

seedlings may regulate mycorrhizal colonization [10, 82] by investing more resources into colonization at high light, where carbon is less limiting [83].

Consistent with expectations, EMF colonization was higher in conspecific than heterospecific soil for *Q. alba* and *Q. rubra*, especially in higher light availability [84, 85], perhaps reflecting the higher specialization of EMF than AMF [81]. For *Q. alba*, EMF colonization also was high in soils cultured by *P. grandidentata* and *Q. rubra*, suggesting association with multiple EMF species.

Phenolics increased with light availability for all study species, but there were no consistent effects of live soil source across all seedling species. Phenolics were higher in conspecific than heterospecific soil for *A. saccharum* across light levels, and for *Q. rubra* at high light. Additionally, phenolics were higher in live than sterilized conspecific soils, suggesting that phenolics increase in the presence of soil-borne microbes. Phenolics production can be induced in response to soil-borne microbes [86, 87], which should be more prevalent in conspecific soil. Although we did not quantify pathogen abundance, we did find that EMF colonization and phenolics were correlated for *Q. alba* and *Q. rubra*. Since EMF colonization was higher in conspecific than heterospecific soil for both species, our results suggest that EMF colonization or the presence of host-specific pathogens induced production of phenolics, especially in conspecific soil.

For EMF-associated species, lignin was higher in conspecific than heterospecific soil, but did not vary with light. Seedling production of lignin may have already reached the upper limit in response to light availability. Seedlings may also achieve greater trait production under less stressful growing conditions, such as in conspecific soils with mutualistic EMF [88]. By improving nutrient availability, EMF can indirectly affect the allocation of seedling resources, potentially impacting lignin synthesis.

NSC increased with light availability for all study species, regardless of soil biota present. This result was consistent with previous studies [89–91], including *Q. alba* [91]. Contrary to expectations, for *Q. alba*, NSC was higher in heterospecific soil. For *P. serotina* and *Q. rubra*, NSC was higher in conspecific soil but only at low light. While one might speculate that seedlings may allocate more NSC to mycorrhizal mutualists or recovery from pathogens in conspecific soil or high light, we did not find strong correlations with NSC for either AMF or EMF colonization.

## Mycorrhizal colonization and functional traits had limited effects on seedling survival

Species differed in which traits, if any, influenced survival. Phenolics, which provide direct chemical defense against soil-borne microbes, could increase survival for *A. saccharum* seedlings and may be the mechanism behind their positive PSFs, supported by greater production of phenolics in conspecific soils [86, 87]. However, *A. saccharum* seedling survival was higher in sterilized than conspecific soil, suggesting that the positive effects of phenolics on survival did not overcome the negative effects of microbes. Furthermore, higher survival in sterilized soil confirms that microbes drove the observed positive PSFs. For *P. serotina*, phenolics were much higher in pooled heterospecific soil and were positively correlated with AMF colonization. *P. serotina* seedlings may be more readily colonized by AMF, regardless of soil source, and thus produce more phenolics in response; this may explain why *P. serotina* are frequently found to have high mortality in conspecific soils [78, 92].

For *Q. alba* seedlings, which are typically EMF-associating, survival decreased as AMF colonization increased. A potential explanation is that AMF can act parasitically in some environmental conditions [48, 93], while EMF provide better direct protection against pathogens.

While it is not well-understood if AMF colonization cause negative PSFs for EMF-associating tree species [94, 95], Bennett et al. [14] found no effect of soil cultured by AMF-associating species on seedlings of EMF-associating species. Furthermore, our pot-based study design may have precluded the benefits of an EMF common mycorrhizal network [96], heightening the negative influence of AMF colonization. Interestingly, *Q. rubra* seedling survival was not influenced by AMF colonization, suggesting that *Q. rubra* may be less reliant upon common mycorrhizal networks or less susceptible to parasitic effects of AMF.

We expected to see traits emerge as stronger drivers of seedling survival, given large intraspecific trait variation in response to soil source and light availability. Although we found that functional traits were influenced by both soil source and light availability, we found limited instances of trait influences on seedling survival. The lack of strong effects could have been driven by stressful field conditions that obscure the importance of functional traits. In contrast to previous greenhouse studies, we did not find PSFs for *A. rubrum*, *P. serotina*, or *Q. rubra* [42, 80]. PSFs quantified in the highly controlled greenhouse conditions often overestimate field measured PSFs [97]. In this study, we took precautions against potential competition and above-ground herbivory from rodents and deer. However, seedlings experienced great variability in rainfall and maximum temperature across the growing season. Heavy rainfall washed away smaller seedlings and would stand in pots if preceded by dryer, warmer periods, which caused soil in the field pots to pull away from the sides of the container and harden. Evaluation of weather data (National Oceanic and Atmospheric Administration) revealed a higher amount of rainfall throughout the field season and a large rainfall event ($> 7$") in July. Variation in weather could have overridden the effects of traits [98] or light availability [99]. PSF experiments carried out in the greenhouse may not detect such environmental effects [100] or may overestimate the strength of PSFs [101].

There are several additional caveats to consider. We expect that there was some contamination of pots via airborne microbes, splash from rain, and falling leaf litter, which may have reduced effect sizes. This might have been seen in the overall low survival rates of both *A. saccharum* and *P. serotina* seedlings. However, we assume that the reported significant effects are the result of treatments, because invasions are random and microbial priority effects should be dominant, especially in whole-soil cores [102]. In addition, the relationship between functional traits and seedling survival is correlative rather than causative since we did not manipulate levels of seedling functional traits. It is also difficult to disentangle some of these trait-survival relationships. For example, while we expect phenolics and lignin to be higher in conspecific soils, and for increases in these traits to lead to higher seedling survival, we also expect higher mortality in conspecific soils where effectively-specialized pathogens are more abundant.

We were unable to separate the impacts of soil-borne mutualists and pathogens on seedling trait values and subsequent survival. Also, we cannot distinguish between direct AMF colonization effects of pathogen reduction through displacement or indirect effects inducing production of phenolics, both of which can enhance seedling survival. Additionally, we were unable to tease apart the effects of colonization type (AMF versus EMF) and seed size, since the EMF-associated species used in this study were large-seeded, and vice-versa (Table 2). Furthermore, this study is limited to four species occurring in a single forest; future studies examining the generalizability of these results should consider additional species and habitats.

## Conclusion

Linking plant traits and environmental conditions to PSFs may help us better understand the role of PSFs in community dynamics [21, 25, 103]. However, most studies have focused on herbaceous plants, which are predominantly colonized by AMF. A focus on tree seedling traits

**Table 2. Local adult abundance, shade tolerance, seed weight, and primary mycorrhizal association for each of our study species.** [1]Local adult abundance was calculated as stems/ha at Alma College's Ecological Preserve; only adults ≥ 5cm dbh were included in this count. [2]Shade tolerance is presented as intolerant, intermediate, or tolerant and as mean ± std. dev., on a standardized scale from 1 (least tolerant) to 5 (most tolerant), calculated by Niinemets and Valladares [56]. [3]Seed weight data was collected from Burns and Honkala [57]. [4]AMF = arbuscular mycorrhizal fungi and EMF = ectomycorrhizal fungi.

| Species | Local adult abundance[1] | Shade tolerance[2] | Seed weight (mg)[3] | Mycorrhizal association[4] |
|---|---|---|---|---|
| Acer rubrum | 131 | Intermediate (3.44 ± 0.23) | 19.7 | AMF |
| Acer saccharum | 285 | very tolerant (4.76 ± 0.11) | 64.9 | AMF |
| Populus grandidentata | 82.33 | Intolerant (1.21 ± 0.27) | 0.2 | AMF & EMF |
| Prunus serotina | 4.33 | Intolerant (2.46 ± 0.34) | 94.3 | AMF |
| Quercus alba | 12.67 | Tolerant (2.85 ± 0.17) | 6,677 | AMF & EMF |
| Quercus rubra | 71.67 | Intermediate (2.75 ± 0.18) | 4,127 | AMF & EMF |

under different environmental conditions, especially in natural field conditions, offers both broader ecological understanding as well as potential applications for forest management. For example, selecting sites with soil and light conditions that promote higher production of defensive and recovery compounds could increase likelihood of seedling restoration success (e.g., *A. saccharum* in conspecific soil). Similarly, it may be beneficial to plant EMF-associating seedlings in soils cultured by other EMF-associating species, to increase potential for positive EMF colonization effects and limit potential negative AMF colonization effects. While environmental conditions could dilute trait effects on seedling survival, in the absence of extreme conditions (as supported by related greenhouse studies), a sharper focus on traits promoting survival rather than growth traits will provide a more mechanistic understanding of forest regeneration dynamics.

## Supporting information

**S1 Fig.** Preliminary boxplots showing the effect of soil core year (2016, 2017) on tree seedling traits: A) AMF colonization, B) EMF colonization, C) Phenolics, D) Lignin, E) NSC. We were concerned that storing soil cores for an extended period of time would have potential negative effects on the microbial community. Specifically, we worried that the soil microbial community would be adversely affected. Preliminary analyses did not indicate any significant effect of soil collection year on seedling trait expression (AMF or EMF colonization, phenolics, lignin, NSC) or survival.
(TIF)

**S2 Fig. Nutrient supply rate (micrograms / 10cm$^2$ / burial length) in seedling pots versus beneath adult trees (Adj.-R$^2$ = 0.92, *p* = 0.03).**
(TIF)

**S3 Fig. Nutrient supply rates (micrograms / 10cm$^2$ / burial length) in sterilized versus live soil, in seedling pots.**
(TIF)

**S4 Fig. Nutrient supply rates (micrograms / 10cm$^2$ / burial length) in each soil source (Acru, Acsa, Prse, Pogr, Qual, Quru) in soil beneath adult trees.** For a) Ca$^{2+}$ and b) Mg$^{2+}$.
(TIF)

**S5 Fig. Light availability in the 18 experimental field plots.** Indirect site factor (ISF, the proportion of diffuse solar radiation at a given location, relative to the amount of diffuse solar radiation in the open) in each subplot (n = 5) per common garden plot (n = 18). For analyses

in which light availability was included as a categorical variable, low = 0.032–0.075 ISF, medium = 0.075–0.118 ISF, and high = 0.118–0.161 ISF.
(TIF)

**S6 Fig. Effect of sterilized versus live soil on seedling traits.** A) AMF colonization, B) EMF colonization, C) phenolics, D) lignin, E) NSC.
(TIF)

**S7 Fig. Effects of light availability and soil source on seedling traits.** A) AMF colonization, B) EMF colonization, C) phenolics, D) lignin, and E) NSC.
(TIF)

**S8 Fig. Correlations between AMF colonization, EMF colonization (for *Q. alba* and *Q. rubra*), phenolics, lignin, and non-structural carbohydrates.**
(TIF)

**S9 Fig. Survival curves for each species, soil source, and light availability.**
(TIF)

**S1 Table. Percent of total soil cores collected in 2016 and 2017.** For each seedling species and soil source. We collected intact soil cores during the summers of 2016 and 2017. Before transplanting the soil cores into the common garden field plots, we stored the cores inside the research field station. Cores were stored after mesh had been glued to the bottom and 2 open sides of the enclosing pots. We covered the open top of the pot with a plastic lid specifically designed to fit our pots. This ensured that there was no potential for contamination of the pots before moving them back to the field.
(TIF)

**S2 Table. Results of t-tests comparing nutrient supply rate (micrograms / 10cm$^2$ / burial length) in seedling pots versus beneath adult trees.** Bolded values are significant at $p < 0.0036$, Bonferroni-corrected to $\alpha = 0.0036$, for original $\alpha = 0.05$ and n = 14 tested nutrients.
(TIF)

**S3 Table. Results of t-tests comparing nutrient supply rates (micrograms / 10cm$^2$ / burial length) in sterilized versus live soil, in seedling pots.** Alpha was Bonferroni-corrected to $\alpha = 0.0038$, for original $\alpha = 0.05$ and n = 13 tested nutrients. *p*-values marked with $^*$ are marginally significant at original $\alpha = 0.05$.
(TIF)

**S4 Table. Full dataset results of t-tests comparing nutrient supply rates (micrograms / 10cm$^2$ / burial length) in sterilized versus live soil, in seedling pots.** For many nutrients, there were not enough replicate pots for a t-test or sample size was small, so we also provide the results of a t-test using the full dataset.
(TIF)

**S5 Table. Results of ANOVAs comparing nutrient supply rates (micrograms / 10cm$^2$ / burial length) in each soil source (Acru, Acsa, Prse, Pogr, Qual, Quru) in soil beneath adult trees.** Adult trees were used as a proxy for seedling pots, since preliminary analyses showed that, for most nutrients, there were no significant differences between nutrient supply rates for soil in seedling pots versus beneath adult trees. For $NO_3^-$, $S^{2+}$, and $Zn^{2+}$, there were significant differences in nutrient supply rate for soil in seedling pots versus adult trees, so we also provide results of ANOVAs for seedling pots. Bolded values are significant at $\alpha = 0.0036$,

Bonferroni-corrected for original α = 0.05 and n = 14 tested nutrients.
(TIF)

**S6 Table. Linear model evaluating the effects of sterilized versus live soil on traits.** AMF colonization, EMF colonization, phenolics, lignin, and NSC. For post-hoc comparisons within species, we used joint tests of estimated marginal means.
(TIF)

**S7 Table. Linear model evaluating the effects of light availability and soil source (conspecific versus pooled heterospecific) on traits (AMF colonization, EMF colonization, phenolics, lignin, and NSC).** Sterilized soil was excluded from the model. For post-hoc comparisons within species, we used joint tests of estimated marginal means.
(TIF)

**S8 Table. Linear model evaluating the effects of light availability and soil source on traits (AMF colonization, EMF colonization, phenolics, lignin, and NSC).** Sterilized soil was excluded from the model. For post-hoc comparisons within species, we used joint tests of estimated marginal means.
(TIF)

**S9 Table. Cox proportional hazards survival models evaluating the effects of light availability and soil source (sterilized versus live conspecific) on seedling survival.** Individual models were performed for each species.
(TIF)

**S10 Table. Cox proportional hazards survival models evaluating the effects of light availability and soil source (conspecific versus pooled heterospecific) on seedling survival.** Individual models were performed for each species.
(TIF)

**S11 Table. Cox proportional hazards survival models evaluating the effects of light availability and soil source (conspecific versus unpooled heterospecific) on seedling survival.** Individual models were performed for each species. Sterilized soil was excluded from the model.
(TIF)

**S1 File. Nutrient availability measurements methodology.**
(DOCX)

**S2 File. Functional trait measurement methodology.**
(DOCX)

## Author Contributions

**Conceptualization:** Katherine E. A. Wood, Richard K. Kobe, Inés Ibáñez, Sarah McCarthy-Neumann.

**Data curation:** Katherine E. A. Wood.

**Formal analysis:** Katherine E. A. Wood, Inés Ibáñez.

**Funding acquisition:** Richard K. Kobe, Sarah McCarthy-Neumann.

**Investigation:** Katherine E. A. Wood, Sarah McCarthy-Neumann.

**Methodology:** Katherine E. A. Wood, Richard K. Kobe, Inés Ibáñez, Sarah McCarthy-Neumann.

**Project administration:** Sarah McCarthy-Neumann.

**Resources:** Richard K. Kobe, Sarah McCarthy-Neumann.

**Supervision:** Sarah McCarthy-Neumann.

**Validation:** Richard K. Kobe, Inés Ibáñez, Sarah McCarthy-Neumann.

**Visualization:** Katherine E. A. Wood.

**Writing – original draft:** Katherine E. A. Wood.

**Writing – review & editing:** Katherine E. A. Wood, Richard K. Kobe, Inés Ibáñez, Sarah McCarthy-Neumann.

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
