## [Decision Letter · Decision Letter 0]

28 Jun 2023

PONE-D-23-14309Tree seedling functional traits mediate plant-soil feedback survival responses across a gradient of light availabilityPLOS ONE

Dear Dr. Wood,

Thank you for submitting your manuscript to PLOS ONE. After careful consideration, we feel that it has merit but does not fully meet PLOS ONE’s publication criteria as it currently stands. Therefore, we invite you to submit a revised version of the manuscript that addresses the points raised during the review process.

The manuscript has received detailed comments from two experts in plant science studies. Both the experts assess that the paper has the potential to be published in PLOS ONE, although they raised some questions about the materials and methods section and results interpretation. Authors need to respond to the point-to-point questions before resubmitting the paper.

We look forward to receiving your revised manuscript.

Kind regards,

Federico Vita

Academic Editor

PLOS ONE

Reviewers' comments:

Reviewer's Responses to Questions

**Comments to the Author**

1. Is the manuscript technically sound, and do the data support the conclusions?

Reviewer #1: Yes

Reviewer #2: Partly

2. Has the statistical analysis been performed appropriately and rigorously? 

Reviewer #1: Yes

Reviewer #2: Yes

3. Have the authors made all data underlying the findings in their manuscript fully available?

Reviewer #1: Yes

Reviewer #2: Yes

4. Is the manuscript presented in an intelligible fashion and written in standard English?

Reviewer #1: Yes

Reviewer #2: Yes

5. Review Comments to the Author

Reviewer #1: Line 40 – What does higher trait values mean, higher phenolics, lignin and NSC? Please be more specific.

Line 43 – Why does this sentence start with ‘however’? Does it relate to the previous sentence somehow? I do not see the connection. Or is it referring to the first sentence of the paragraph?

Line 56 – What is “This”, vague sentence topic.

Like 84 – AMF and EMF “can” increase their host plant’s… They do not always as in a beneficial way.

Line 107-108 – How does low light availability lead to increased pathogen abundance?

Line 147-148 - This could also suggest highly virulent pathogens. Were those seedlings inspected for root rot or damping off?

Line 201- Were light measurements taken under hardware cloth or above the hardware cloth? 2.5 mm mesh can create significant shading.

Line 244 – How were these light thresholds chosen?

Line 364 – In the end they all died though…Is it biologically significant that they lasted a week or two longer in sterilized soil? I find it strange that all the sugar maple died.

Line 377 – the word cultured does not feel right when used with abiotic variables, I think it should be reserved for biotic interactions. Consider changing it to ‘modified’

Line 384 – The citations here could use more context. Why not discuss what those two studies found and how it relates to your findings? Have others found higher AMF colonization in conspecific soils in other PSF experiments?

Line 408-410 This paragraph could use some support from the literature.

Reviewer #2: Review of PONE-D-23-14309 “Tree seedling functional traits mediate plant-soil feedback survival responses across a gradient of light availability”

This manuscript describes a field study that tests how light availability, plant traits (including lignin, phenolics, NSC, and mycorrhizal colonization), and soil source (conspecific, heterospecific, or sterilized) manifest and affect seedling survival in 4 common temperate tree seedling species. A major strength of this study is its novel and careful experimental design: seedlings were planted in the field using intact soil cores, and therefore the results are much more likely to reflect reality than similar studies that have taken place in greenhouses. I think the is an excellent piece of work that contributes to the growing body of literature on temperate forest PSF and plant traits, and applaud the authors for the design and scale of the study! I have one overall concern and some minor comments, below:

My major concern has to do with the interpretation of the PSF results (e..g mainly figure 1a). Firstly, it seems that for both the A. saccharum and P. serotina, all seedlings died before the end of the growing season. This makes the interpretation of any apparent PSF effects in these species somewhat difficult, as some seedlings need to survive for any PSF effects to have community level consequences. I see how the authors approach of using survival curves, rather than overall survival at the end of the season, could make sense (particularly in light of the observation that the growing season was atypically dry, and therefore perhaps overall survival was depressed), but I think this should be more prominently discussed in the text of the manuscript: ideally the overall survival should be reported for each species (and perhaps by group, as again, that is a biologically important measure), as well as how the high mortality on those species influence the interpretation of the results.

Relatedly, as I point out below, I don’t fully understand the authors interpretation of the Q. rubra results as presented in Fig 1a. Again this is based only on eyeballing the figure, but it seems that about 50% of Q. rubra in conspecific soil survived, compared to 60-65% in heterospecific soil and >75% in sterilized soil. This (to me) was a striking difference, but it was not mentioned in the results text, which I took to mean that the differences were not statically significant in the models that were run. This was particularly confusing as, based on the figure, a reader comes to the conclusion that the biggest differences among soil sources are found in Q. rubra seedlings, but the results suggest that the only differences are found in A. saccharum (for which the figure makes the soil sources look virtually identical). It would seem to me that, regardless of model structure or approach, an overall growing season difference in survival of that scale would be biologically important. I would suggest that the authors compare their model results to the figures, and for any major discrepancies (such as with Q. rubra, I would argue) try and determine and report the cause of the difference in the text (as I suspect I won’t be the only reader with this question!). Regardless, I think that reporting the survival at the end of the season by species and group (as suggested above), would be a simple addition that would also help provide more context to the survival curve analysis.

Line comments:

Line 40: “higher trait values” isn’t very descriptive: could this be replaced by an example of a trait? (e.g. higher colonization, lignin, etc)

Line 61: define AMF

Line 91: is there any reason to think (e.g. plausible mechanism by which) that soil microbial community (or soil chemistry, which also may differ beneath con- and heterospecifics?) might influence lignin production?

Line 103-104: I would find a conceptual diagram *extremely* helpful for this paper. It would not need to be a piece of art: even a simple box and arrow diagram would do. However there are sveral complex, interacting factors being tested here: light availability, NSC and lignin, mycorrhizal colonization, and seedling success. A visual aid that clearly identifies the putative interactions would help immensely: as it, the introduction is a bit hard to follow as so many topics are introduced.

Line 107, 11, 115: “This supports that...” confusing wording.

Line 114-124: there appears to be some circular logic happening in hypothesis 3 vs 5: lignin and phenolics are higher due to conspecific soil (which ought to decrease survival), but higher lignin and phenolics should lead to higher survival. Without a time series, it would be impossible to disentangle these.

Line 156: should “long” be “deep”?

Line 166: could local abundance be presented either in stems/hectare or Ba/hectare to make it more comparable/understandable?

Line 230: When you say “We compared amounts of measured traits...” do you mean this was an a priori linear contrast? Or what do you mean by compare?

Line 243; does “ISF” have any units? And how is it calculated?

Line 255-256 Can you provide more details about the imputation? What factors was is based upon?

Line 262 This result seems somewhat at odds with Fig 1A: it looks like 100% of the A. saccharum seedlings died (and at almost identical time horizons), regardless of soil origin. In contrast, the oaks seems to have very different survival curves based on soil origin, but apparently no “significant” differences. I would encourage you to rethink the statistical test that is being used to produced these results: to me, it seems far more biologically relevant that ~10% (eyeballing here) more Q. rubra survived in heterospecific soil than in conspecific soil at day 112 than whatever statistical difference exists between the A. saccharum curves, which nonetheless result in complete mortality in all treatments by day 70. If this is somehow a result of the model structure being somewhat different than the visualization, I would explain that clearly or I think other readers will also be left confused.

Line 287 Fig 2 is an *awesome* figure, and this is minor, but I find the empty Acer and Prunus panels in the second row distracting

Line 296 (and throughout): The double negative of “non-sterilized” takes me a minute each time to get my head around and makes interpreting the comparisons clunky. Could this be replaced with another term (e.g. “fresh” soil or “live” soil) to make this easier?

Line 348 *removed

Line 250: I’m still not convinced I agree with this interpretation: Q. rubra survival looks to me like reasonably strong evidence of negative PSF. Even if the model predictions are not significant at p = 0.05, I would certainly interpret this as possible negative PSF, as the community level effects of that big of a difference in groups (conspecific vs pooled heterospecific) in ultimate survival are not trivial.

Line 362 could you provide any data on how dry (relative so say a 30-year climate average) this growing season was, to give a sense of how likely this was a factor?

Line 382 Wouldn’t this depend on the target species? In other words, I would expect AMF colonization to be higher in conspecifics for the AM-seedling species, but lower for the ECM-seedling species.

Line 398 Is this not contradictory? “but there were no consistent effects of soil source. Phenolics were higher in non-sterilized than sterilized conspecific soils”? Please clarify.

Line 408-410 Could you provide a bit more discussion of/context for these results?

Line 480-481 Again, I believe this runs contrary to your results (that Q. rubra survival was lowest in fresh conspecific soil)

6. PLOS authors have the option to publish the peer review history of their article (what does this mean?). If published, this will include your full peer review and any attached files.

Reviewer #1: No

Reviewer #2: No

---

## [Author Response · Author response to Decision Letter 0]

12 Aug 2023

1. Is the manuscript technically sound, and do the data support the conclusions?

Reviewer #1: Yes

Reviewer #2: Partly

2. Has the statistical analysis been performed appropriately and rigorously?

Reviewer #1: Yes

Reviewer #2: Yes

3. Have the authors made all data underlying the findings in their manuscript fully available?

Reviewer #1: Yes

Reviewer #2: Yes

4. Is the manuscript presented in an intelligible fashion and written in standard English?

Reviewer #1: Yes

Reviewer #2: Yes

5. Review Comments to the Author

Reviewer #1:

1. Line 40 – What does higher trait values mean, higher phenolics, lignin and NSC? Please be more specific.

To be more specific, we added a parenthetical statement “higher trait values (measured amounts of a given trait)”. 

2. Line 43 – Why does this sentence start with ‘however’? Does it relate to the previous sentence somehow? I do not see the connection. Or is it referring to the first sentence of the paragraph?

We removed the word “however” from this sentence.

3. Line 56 – What is “This”, vague sentence topic.

We changed the word “This” to “These feedbacks” to better relate back to the previous sentence.

4. Like 84 – AMF and EMF “can” increase their host plant’s… They do not always as in a beneficial way.

The reviewer is correct that AMF and EMF do not always act beneficially. We added “can”. 

5. Line 107-108 – How does low light availability lead to increased pathogen abundance?

This is in reference to line 98: “Higher mortality from pathogens typically occurs in low light [20,40], where wetter and cooler conditions enhance microbe reproduction and dispersal [13,43,47].”

6. Line 147-148 - This could also suggest highly virulent pathogens. Were those seedlings inspected for root rot or damping off?

During each census, we scored seedlings for health category and noted damping-off symptoms, but dead roots were not inspected for rot nor any other quantifiable pathogen data collected. Seedling at this time were so small that, even within 3-4 days between censuses, we didn't have much tissue to tell the cause of death.

7. Line 201- Were light measurements taken under hardware cloth or above the hardware cloth? 2.5 mm mesh can create significant shading.

We revised this paragraph to clarify the types of hardware cloth we used and their potential limitations on seedlings. We also added this sentence: “Seedlings likely did not experience significant shading due to the addition of the hardware cloth and often grew above the cloth within 2 weeks of planting.”

8. Line 244 – How were these light thresholds chosen?

We added the sentence: “These light thresholds were determined by taking the range of light availability across the field plots and equally dividing into three bins.”

9. Line 364 – In the end they all died though…Is it biologically significant that they lasted a week or two longer in sterilized soil? I find it strange that all the sugar maple died.

To also address the total amount of survival, not just the rates of survival, we made several revisions throughout the manuscript and to the figures. See detailed revisions under responses to Reviewer 2 (below). 

10. Line 377 – the word cultured does not feel right when used with abiotic variables, I think it should be reserved for biotic interactions. Consider changing it to ‘modified’.

We changed the word “cultured” to “modified”, better reflecting an influence of the adult tree on abiotic factors.

11. Line 384 – The citations here could use more context. Why not discuss what those two studies found and how it relates to your findings? Have others found higher AMF colonization in conspecific soils in other PSF experiments?

We expanded upon references in this paragraph, including Bennett et al. 2017, who found higher AMF colonization in conspecific soils. 

12. Line 408-410 This paragraph could use some support from the literature.

We have expanded upon this paragraph and added supporting references [85, 86].

 

Reviewer #2: 

Review of PONE-D-23-14309 “Tree seedling functional traits mediate plant-soil feedback survival responses across a gradient of light availability”

This manuscript describes a field study that tests how light availability, plant traits (including lignin, phenolics, NSC, and mycorrhizal colonization), and soil source (conspecific, heterospecific, or sterilized) manifest and affect seedling survival in 4 common temperate tree seedling species. A major strength of this study is its novel and careful experimental design: seedlings were planted in the field using intact soil cores, and therefore the results are much more likely to reflect reality than similar studies that have taken place in greenhouses. I think the is an excellent piece of work that contributes to the growing body of literature on temperate forest PSF and plant traits, and applaud the authors for the design and scale of the study! I have one overall concern and some minor comments, below:

1. My major concern has to do with the interpretation of the PSF results (e.g. mainly figure 1a). Firstly, it seems that for both the A. saccharum and P. serotina, all seedlings died before the end of the growing season. This makes the interpretation of any apparent PSF effects in these species somewhat difficult, as some seedlings need to survive for any PSF effects to have community level consequences. I see how the authors approach of using survival curves, rather than overall survival at the end of the season, could make sense (particularly in light of the observation that the growing season was atypically dry, and therefore perhaps overall survival was depressed), but I think this should be more prominently discussed in the text of the manuscript: ideally the overall survival should be reported for each species (and perhaps by group, as again, that is a biologically important measure), as well as how the high mortality on those species influence the interpretation of the results.

First, we noticed an error with our original submission of Figure 1a, which was a repeat of Figure 1b.

We feel that the figures now sufficiently represent the survivorship differences across the growing season. We have uploaded the corrected and updated figures to this submission and also include them embedded below. (Note that Fig 1 is now Fig 2, after the addition of a conceptual diagram to the introduction).

 

Figure 2a

 

Figure 2b

  

2. Relatedly, as I point out below, I don’t fully understand the authors interpretation of the Q. rubra results as presented in Fig 1a. Again this is based only on eyeballing the figure, but it seems that about 50% of Q. rubra in conspecific soil survived, compared to 60-65% in heterospecific soil and >75% in sterilized soil. This (to me) was a striking difference, but it was not mentioned in the results text, which I took to mean that the differences were not statically significant in the models that were run. This was particularly confusing as, based on the figure, a reader comes to the conclusion that the biggest differences among soil sources are found in Q. rubra seedlings, but the results suggest that the only differences are found in A. saccharum (for which the figure makes the soil sources look virtually identical). It would seem to me that, regardless of model structure or approach, an overall growing season difference in survival of that scale would be biologically important. I would suggest that the authors compare their model results to the figures, and for any major discrepancies (such as with Q. rubra, I would argue) try and determine and report the cause of the difference in the text (as I suspect I won’t be the only reader with this question!). Regardless, I think that reporting the survival at the end of the season by species and group (as suggested above), would be a simple addition that would also help provide more context to the survival curve analysis.

The Q. rubra survival values in the original figure are incorrect; they are the light value survival, not the soil survival (see above corrected figures). Per the reviewer’s suggestions, we added discussion of these results to the discussion caveats section.

Line comments:

3. Line 40: “higher trait values” isn’t very descriptive: could this be replaced by an example of a trait? (e.g. higher colonization, lignin, etc)

To be more specific, we added a parenthetical statement “higher trait values (measured amounts of a given trait)”. 

4. Line 61: define AMF

Corrected to “Arbuscular mycorrhizal fungi (AMF)”.

5. Line 91: is there any reason to think (e.g. plausible mechanism by which) that soil microbial community (or soil chemistry, which also may differ beneath con- and heterospecifics?) might influence lignin production?

We added “However, it could be driven by soil nutrient availability, which can be impacted by microbes [37,38]” and supporting references.

6. Line 103-104: I would find a conceptual diagram *extremely* helpful for this paper. It would not need to be a piece of art: even a simple box and arrow diagram would do. However there are several complex, interacting factors being tested here: light availability, NSC and lignin, mycorrhizal colonization, and seedling success. A visual aid that clearly identifies the putative interactions would help immensely: as it, the introduction is a bit hard to follow as so many topics are introduced.

We added a conceptual diagram to the introduction, before the hypotheses (also included below).

Fig 1. Conceptual diagram demonstrating the relationships between light availability, functional traits (phenolics, lignin, and nonstructural carbohydrates [NSC]), colonization by mycorrhizal fungi, and tree seedling survival. Green, solid lines indicate a proposed positive relationship. Red, dashed lines indicate a proposed negative relationship. Lines that directly influence tree seedling survival are thicker. Stars (*) next to the lines linking ‘Mycorrhizae’ with ‘NSC’ and ‘Seedling Survival’ indicate that this relationship is usually positive but can shift to neutral or negative.

7. Line 107, 11, 115: “This supports that...” confusing wording.

We changed the wording to “This result would indicate”. 

8. Line 114-124: there appears to be some circular logic happening in hypothesis 3 vs 5: lignin and phenolics are higher due to conspecific soil (which ought to decrease survival), but higher lignin and phenolics should lead to higher survival. Without a time series, it would be impossible to disentangle these.

We have added this caveat to the discussion, addressing the difficulty of disentangling these interactions. “It is also difficult to disentangle some of these trait-survival relationships. For example, while we expect phenolics and lignin to be higher in conspecific soils, and for increases in these traits to lead to higher seedling survival, we also expect higher mortality in conspecific soils where effectively-specialized pathogens are more abundant.”

9. Line 156: should “long” be “deep”?

Yes, we changed “long” to “deep”.

10. Line 166: could local abundance be presented either in stems/hectare or Ba/hectare to make it more comparable/understandable?

We altered the wording to “stems/ha” and included a caveat that “only adults ≥ 5 cm dbh were included in this count”.

11. Line 230: When you say “We compared amounts of measured traits...” do you mean this was an a priori linear contrast? Or what do you mean by compare?

Yes. To improve clarity, we added the language “We used a priori linear contrasts to compare”. 

12. Line 243; does “ISF” have any units? And how is it calculated? 

We added this information to the methods: “ISF represents the proportion of diffuse (indirect) solar radiation reaching a given location, relative to an open site and was calculated using HemiView software (Delta-T Devices, Ltd., Burwell, England).”

13. Line 255-256 Can you provide more details about the imputation? What factors was is based upon?

We added this sentence to clarify that imputed colonization and trait data were imputed “from seedlings harvested at three weeks, for each combination of seedling species, plot, soil source, and light level.”

14. Line 262 This result seems somewhat at odds with Fig 1A: it looks like 100% of the A. saccharum seedlings died (and at almost identical time horizons), regardless of soil origin. In contrast, the oaks seems to have very different survival curves based on soil origin, but apparently no “significant” differences. I would encourage you to rethink the statistical test that is being used to produced these results: to me, it seems far more biologically relevant that ~10% (eyeballing here) more Q. rubra survived in heterospecific soil than in conspecific soil at day 112 than whatever statistical difference exists between the A. saccharum curves, which nonetheless result in complete mortality in all treatments by day 70. If this is somehow a result of the model structure being somewhat different than the visualization, I would explain that clearly or I think other readers will also be left confused.

Please see above comments about the revised Fig 1a. Per the reviewer’s suggestions, we added tables to each of the figures that show the total number of surviving seedlings (number at risk), so there is an alternative way to interpret the results, beyond the survival curves. We also added discussion of these results to the discussion caveats section.

15. Line 287 Fig 2 is an *awesome* figure, and this is minor, but I find the empty Acer and Prunus panels in the second row distracting.

We have removed the empty panels from Fig 2 (now Fig 3).

16. Line 296 (and throughout): The double negative of “non-sterilized” takes me a minute each time to get my head around and makes interpreting the comparisons clunky. Could this be replaced with another term (e.g. “fresh” soil or “live” soil) to make this easier?

We have changed instances of “non-sterilized” to “live” soils.

 

17. Line 348 *removed

Fixed.

18. Line 250: I’m still not convinced I agree with this interpretation: Q. rubra survival looks to me like reasonably strong evidence of negative PSF. Even if the model predictions are not significant at p = 0.05, I would certainly interpret this as possible negative PSF, as the community level effects of that big of a difference in groups (conspecific vs pooled heterospecific) in ultimate survival are not trivial.

Please see above comments about the revised Fig 1a. Per

19. Line 362 could you provide any data on how dry (relative so say a 30-year climate average) this growing season was, to give a sense of how likely this was a factor?

We evaluated the NOAA weather data for the 5 years on either end of this field season. We could not find evidence of a dryer overall field season, but the temperature experienced was still dry enough to cause soil to pull from the sides of the field pots. Additionally, we did identify an abnormal amount of rainfall, including a 7” event in the middle of July. We added this information to the discussion. 

20. Line 382 Wouldn’t this depend on the target species? In other words, I would expect AMF colonization to be higher in conspecifics for the AM-seedling species, but lower for the ECM-seedling species.

We removed “Surprisingly,” from this sentence and expanded on the reference cited here.

21. Line 398 Is this not contradictory? “but there were no consistent effects of soil source. Phenolics were higher in non-sterilized than sterilized conspecific soils”? Please clarify.

We changed this sentence to “there were no consistent effects of live soil across all seedling species”. We also reordered the subsequent sentences to increase clarity. 

22. Line 408-410 Could you provide a bit more discussion of/context for these results? 

We added several sentences to expand upon these results.

23. Line 480-481 Again, I believe this runs contrary to your results (that Q. rubra survival was lowest in fresh conspecific soil)

While there is lower survival of Quercus spp. in live conspecific relative to heterospecific soils, there was no apparent negative biotic effect indicated by comparisons of live versus sterilized conspecific soils. There may be abiotic factors influencing seedling survival for these species. We still think it may be beneficial to plant Quercus spp. seedlings in soils with EMF, due to the increased availability of resources and higher measured trait values.

---

## [Decision Letter · Decision Letter 1]

27 Sep 2023

PONE-D-23-14309R1Tree seedling functional traits mediate plant-soil feedback survival responses across a gradient of light availabilityPLOS ONE

Dear Dr. Wood,

Thank you for submitting your manuscript to PLOS ONE. After careful consideration, we feel that it has merit but does not fully meet PLOS ONE’s publication criteria as it currently stands. Therefore, we invite you to submit a revised version of the manuscript that addresses the points raised during the review process.

We look forward to receiving your revised manuscript.

Kind regards,

Federico Vita

Academic Editor

PLOS ONE

Journal Requirements:

Additional Editor Comments:

The reviewers carefully checked the revised version of the paper. There is still a minor issue that authors need to address before being able to accept the article for publication. Please see the reviewer's comments for more details.

Reviewers' comments:

Reviewer's Responses to Questions

**Comments to the Author**

1. If the authors have adequately addressed your comments raised in a previous round of review and you feel that this manuscript is now acceptable for publication, you may indicate that here to bypass the “Comments to the Author” section, enter your conflict of interest statement in the “Confidential to Editor” section, and submit your "Accept" recommendation.

Reviewer #1: All comments have been addressed

Reviewer #2: (No Response)

2. Is the manuscript technically sound, and do the data support the conclusions?

Reviewer #1: Yes

Reviewer #2: Yes

3. Has the statistical analysis been performed appropriately and rigorously? 

Reviewer #1: Yes

Reviewer #2: Yes

4. Have the authors made all data underlying the findings in their manuscript fully available?

Reviewer #1: Yes

Reviewer #2: Yes

5. Is the manuscript presented in an intelligible fashion and written in standard English?

Reviewer #1: Yes

Reviewer #2: Yes

6. Review Comments to the Author

Reviewer #1: All of my concerns were adequately addressed. All of my concerns were adequately addressed. All of my concerns were adequately addressed.

Reviewer #2: Although the authors have addressed one of my two initial concerns about the manuscript by updating their figures (and including a conceptual diagram: that is very helpful!), my concern regarding the presentation of overall survival results remains. In their response, the authors indicated that on the new versions of the figures there would be an indication of the group level survival; however, in the version I received there is still no reporting or discussion of the group level, overall survival. This is a crucial result that cannot be omitted; it is necessary for the interpretation of all other results. The authors include language like "although the majority of A. saccharum and P. serotina died by the end of the

487 growing season" (line 486), however, it seems from the figures that 100% of these seedlings died. I encourage the authors to simply include a table that states the overall survival at the end of the season of each species in each group (soil source and light), and to directly refer to this in the results and discussion. For example, if none of the A. saccharum seedlings survived, this must be noted, as it weakens (but doesn't completely discount) the assertion that this species exhibited PSF.

7. PLOS authors have the option to publish the peer review history of their article (what does this mean?). If published, this will include your full peer review and any attached files.

Reviewer #1: No

Reviewer #2: No

---

## [Author Response · Author response to Decision Letter 1]

9 Oct 2023

We thank the reviewers for their feedback. We have added a new table (Table 2, Line 288) that states the overall survival (number of seedlings and percentage) at the end of the season for each species in each group (soil source and light). Because there was not a significant interaction effect between soil source and light availability on seedling survival, we presented the results for soil source and light availability separately. Table section A aligns with figure 2A, and table section B aligns with figure 2B. In addition, we added a paragraph about these results to the discussion (Line 398).

---

## [Editor Report · Decision Letter 2]

23 Oct 2023

Tree seedling functional traits mediate plant-soil feedback survival responses across a gradient of light availability

PONE-D-23-14309R2

Dear Dr. Wood,

We’re pleased to inform you that your manuscript has been judged scientifically suitable for publication and will be formally accepted for publication once it meets all outstanding technical requirements.

Kind regards,

Federico Vita

Academic Editor

PLOS ONE

Additional Editor Comments (optional):

The authors addressed all the questionable points raised in the previous rounds of revision. The current version of the paper has been improved, and it reached the quality level for publication in Plos One.
---

## [Editor Report · Acceptance letter]

15 Nov 2023

PONE-D-23-14309R2 

Tree seedling functional traits mediate plant-soil feedback survival responses across a gradient of light availability 

Dear Dr. Wood:

I'm pleased to inform you that your manuscript has been deemed suitable for publication in PLOS ONE. Congratulations! Your manuscript is now with our production department. 

Kind regards, 

on behalf of

Dr. Federico Vita 

Academic Editor

PLOS ONE